# Learning important common data elements from shared study data: The All of Us program analysis

Craig S. Mayer◉*◉, Vojtech Huser◉

Lister Hill National Center for Biomedical Communication, National Library of Medicine, NIH, Bethesda, Maryland, United States of America

◉ These authors contributed equally to this work.
* craig.mayer2@nih.gov

**Data Availability Statement:** The data is available at our project repository https://github.com/lhncbc/CRI/tree/master/AoU/CDE. This includes all analytic code and aggregated data that was analyzed.

## Abstract

There are many initiatives attempting to harmonize data collection across human clinical studies using common data elements (CDEs). The increased use of CDEs in large prior studies can guide researchers planning new studies. For that purpose, we analyzed the All of Us (AoU) program, an ongoing US study intending to enroll one million participants and serve as a platform for numerous observational analyses. AoU adopted the OMOP Common Data Model to standardize both research (Case Report Form [CRF]) and real-world (imported from Electronic Health Records [EHRs]) data. AoU standardized specific data elements and values by including CDEs from terminologies such as LOINC and SNOMED CT. For this study, we defined all elements from established terminologies as CDEs and all custom concepts created in the Participant Provided Information (PPI) terminology as unique data elements (UDEs). We found 1 033 research elements, 4 592 element-value combinations and 932 distinct values. Most elements were UDEs (869, 84.1%), while most CDEs were from LOINC (103 elements, 10.0%) or SNOMED CT (60, 5.8%). Of the LOINC CDEs, 87 (53.1% of 164 CDEs) originated from previous data collection initiatives, such as PhenX (17 CDEs) and PROMIS (15 CDEs). On a CRF level, The Basics (12 of 21 elements, 57.1%) and Lifestyle (10 of 14, 71.4%) were the only CRFs with multiple CDEs. On a value level, 61.7% of distinct values are from an established terminology. AoU demonstrates the use of the OMOP model for integrating research and routine healthcare data (64 elements in both contexts), which allows for monitoring lifestyle and health changes outside the research setting. The increased inclusion of CDEs in large studies (like AoU) is important in facilitating the use of existing tools and improving the ease of understanding and analyzing the data collected, which is more challenging when using study specific formats.

## 1 Introduction

Common Data Elements (CDEs) represent an effort to standardize data collection across human clinical studies [1]. There are many CDE initiatives and the total number of defined CDEs can be overwhelming. For example, Huser et al. found 1 414 PhenX data elements across

**Funding:** This project was funded as part of the Intramural research program at the National Library of Medicine. The funders had no role in study design, data collection and analysis, decision to publish, or preparation of the manuscript.

**Competing interests:** NO authors have competing interests.

426 studies deposited in dbGaP [2]. For principal investigators (PIs) or Study Coordinators, picking which CDEs to adopt during study design can be a challenging task, which can be mitigated by relying on CDEs adopted by existing, large cohort, observational studies (as a guide). The use of such CDEs across multiple studies also allows for the ability to quickly perform analyses across multiple data collection efforts, as well as allow for the reuse of preexisting analytic tools.

We review data from the "All of Us" (AoU) program [3] in the United States that plans to enroll one million participants and serve as a platform for numerous observational research analyses. In this study, we identify the CDEs used, the usage of all data elements and permissible values in the program, as well as the originating source of the data elements. The identification of which CDEs are used compared to when unique data elements (UDEs) are developed in a large multi-use study, like AoU, can serve as lessons for future study design and CDE integration.

## 2 Materials

### 2.1 All of Us program

The AoU program began nationwide enrollment in May 2018. The total funding allocated to the program is $2.13 billion [4]. The aim of the AoU program is to develop a diverse, information rich database that serves as a central point for many secondary research studies and reduce the need for developing individual single use study specific data collection protocols. During the study design stage, the AoU team carefully considered elements to include in a selection of Case Report Forms (CRFs). Organizers focused on utilizing existing standardized data collection instruments, such as Population Assessment of Tobacco and Health Study (PATH), AUDIT-C (Alcohol Use Disorders Identification Test-Concise), Behavioral Risk Factor Surveillance System (BRFSS), and National Health Interview Survey (NHIS). Many elements from such initiatives were included in the program's surveys. During enrollment in the AoU program, each participant completes three core survey CRFs titled The Basics, Overall Health, and Lifestyle. In addition to core CRFs, they may complete additional surveys to share their experiences with health care access and their personal and family medical history. Additional surveys have also been added after the start of the program, such as the COVID-19 Participant Experience (COPE) survey which was launched in May 2020. In addition to participant provided information via CRFs, participants can consent to have data imported from their Electronic Health Record (EHR). AoU combines the CRF data collected during a research visit with EHR data generated during routine healthcare visits that occurred both during AoU enrollment, and prior to the start of the program. Continuous import of EHR data provides a cost-effective participant follow-up and classifies AoU as a program that combines research-specific data collection with Real World Data (RWD). Along with the provided survey results and imported EHR data, AoU has incorporated other types of data and continues to enrich the database as the program grows. This includes both genetic and wearable device data. While data pertaining to genetics and wearable devices are present in the workbench, we do not analyze them as part of this study as they are not conformed to the same data model as the survey and EHR data.

### 2.2 Data analysis workbench

AoU allows access to de-identified individual participant data for interested researchers via a cloud-based research workbench platform. Researchers are issued login credentials and they are required to use a separate workspace for each research project. Research users must state the goal of their research projects and they are posted on the AoU public

website. There are two tiers of individual participant data access (controlled and registered) with a differing level of data de-identification and redaction between the two tiers. The *controlled* tier of access has minimal data redaction and was made available to researchers in June 2022, while the *registered* tier with more data redaction has been available since 2020 and used by our study. Finally, unlike the previous two tiers offering individual participant data, the *public tier* offers aggregated participant data accessible without login. The workbench supports two programming languages: R and Python, and data is stored in a Google BigQuery database platform that supports Structured Query Language (SQL) queries.

## 2.3 Data model

To standardize the study's data, AoU adopted the Observational Medical Outcomes Partnership (OMOP) Common Data Model (CDM) [5]. The OMOP model uses concepts to organize data. OMOP acknowledges that there are many different terminologies and tries to designate a preferred terminology for certain data domains (e.g., diagnostic history or medication history). Terminologies are organized in the OMOP CONCEPT table. Each concept belongs to a single vocabulary (or terminology). Each vocabulary is identified by a string identifier called the vocabulary_id. For example, the vocabulary_id of the LOINC terminology is 'LOINC'. Per OMOP model specifications, an OMOP concept can be of two types: standard (preferred concepts that are used to represent data) or non-standard (non-preferred concepts that are instead mapped to standard concepts).

While AoU mapped many data elements and permissible value concepts to established terminologies (such as the Logical Observation Identifiers Names and Codes [LOINC] or Systematic Nomenclature of Medicine—Clinical Terms [SNOMED CT]) [6], the study also created new custom concepts (5 743 overall in the CONCEPT table). These custom concepts are maintained in a custom terminology (referred to as 'PPI'). The PPI (Participant Provided Information) terminology is available in the OMOP model vocabulary layer, sometimes referred to as Athena, and concepts in the terminology are thus available to other studies adopting the OMOP model [7].

## 3 Methods

To conduct our analysis, we used the AoU workbench, and the data released for the registered tier in the third quarter of 2021 (referenced by AoU as R2021Q3R2 release). AoU policy restricts the extraction of data from the workbench to only aggregated data. For privacy reasons, only aggregated data that combines data from at least 20 participants are allowed to be exported. In compliance with this policy, we do not include concepts that do not meet this requirement in any table or supplemental file.

In a prior clinical research informatics publication [8], we defined the terms *data element dictionary* (list of all data elements used for data collection in a study) and *permissible value dictionary* (list of all permissible values used as values of categorical data elements). We analyze AoU data on these two levels of data elements (elements) and permissible values (values). We used R and SQL language to query AoU data in the workbench. We take advantage of an R package (developed by our team and available at our GitHub repository in the following reference) called r4aou [9] that helps with common analytical tasks within the AoU workbench, and includes functions that return SQL query results, writes files for extraction, runs previously established cohorts, etc.

### 3.1 Data element level

A fact about a patient can be represented in the OMOP model in two data tables: (1) the MEASUREMENT table, which is used for results of a standardized test or some other activity that generates a quantitative or qualitative result; and (2) the OBSERVATION table, which is used for additional facts obtained in the context of clinical examination, survey questioning, or a procedure not included in measurements.

For this study, we use the term data element (element) to represent both measurements and observations. We analyzed the AoU data for the number of distinct elements used, their overall usage volume, the percent of participants with data corresponding to each element, and the terminology source for the element (OMOP's vocabulary_id).

Looking at the vocabulary_id (terminology) of each element, we declare each element as a CDE or a UDE. We define CDEs as elements that use one of the established standard terminologies, such as LOINC or SNOMED CT. In contrast, we define UDEs as elements for which the AoU team did not find a suitable concept in an existing healthcare terminology or CDE initiative and created a new concept in their custom PPI terminology.

Furthermore, we determined the data type of each element by stating whether an element was quantitative or qualitative. The OMOP model allows each data row to have two value columns: value_as_number (data type: numeric) and value_as_concept_id (data type: coded concept). The OMOP model specification is ambiguous as to whether the values are mutually exclusive. It is also important to note that an element use may be variable across data collection sites. We classified an element as a numeric type if more than half the data rows for the concept had a value in the value_as_number column.

We also identified which CDEs were from a previously created data collection instrument (or initiative CDEs). Such initiatives include Patient-Reported Outcomes Measurement Information System (PROMIS) and PhenX. We identified which elements are from such initiatives by joining the method type from the LOINC table to the concept code (LOINC Code). These initiatives showcase the adoption of CDEs across multiple data collection initiatives.

To Identify which elements were from CRFs (as opposed to EHR) we used the data source, or src_id, which states what site a data row is generated from. If the src_id is listed as 'PPI/PM' then the data row originates from a CRF or research visit. We consider the research visit as part of the CRF data since it is data collected as part of the research protocol and is obtained as part of regular research processes (instead of imported via EHR). For brevity, we report only on the research CRF elements in this paper, but information regarding elements imported from EHR can be found at our study repository [10].

Using the CONCEPT_RELATIONSHIP table we created a CRF data dictionary that assigns each element to the topic and CRF that the element originated from. We counted how many elements and CDEs are present from each CRF.

### 3.2 Permissible value level

For categorical data elements or qualitative tests, the standardization does not end at the element level (e.g., marital status as a question) but must further continue to the standardization of the values, or all permitted answer options (e.g., divorced as an answer). Table 1 shows an example of standardized permissible values for the question of marital status.

Expressing the overall hierarchy, Fig 1 shows a representation of the overall flow of CRF, topic, data element and permissible value using the example data element of 'How often do you have a drink containing alcohol'.

Fig 1 shows how a CRF (Lifestyle) subsumes a topic (Alcohol), which contains a data element ('How often do you have a drink containing alcohol') which has permissible values

**Table 1. Example of a categorical element with standardized permissible values (marital status).**

| Data Element Level | | | | Permissible Value Level | | | | |
|---|---|---|---|---|---|---|---|---|
| **Concept ID** | **Concept Name** | **Vocabulary ID** | **Concept Code** | **Concept ID** | **Concept Name** | **Vocabulary ID** | **Concept Code** | **Frequency** |
| 3046344 | Marital status | LOINC | 45404–1 | 45876756 | Married | LOINC | LA48-4 | 41.22% |
| | | | | 45881671 | Never married | LOINC | LA47-6 | 26.24% |
| | | | | 45883375 | Divorced | LOINC | LA51-8 | 14.20% |
| | | | | 45883710 | Living with partner | LOINC | LA15605-1 | 6.65% |
| | | | | 45883711 | Widowed | LOINC | LA49-2 | 5.21% |
| | | | | 45884459 | Separated | LOINC | LA4288-2 | 3.63% |
| | | | | 1177221 | I prefer not to answer | LOINC | LA29631-1 | 1.99% |
| | | | | 903096 | PMI: Skip | PPI | PMI_Skip | 0.87% |

('Never' or '2–3 times a week'). Fig 1 also shows the flow of imported EHR data into such data elements, which will be discussed later.

On the value level, we analyzed the use of values (1) throughout the dataset regardless of element, and (2) grouped by element. On the full dataset level, since a value can be used with multiple elements, for each value we quantified the volume of usage and identified the most frequently used values. We also, for each element, analyzed different values used, which included reviewing how often a value appeared for each element (most common values), how many distinct values were used for a given element, and the terminology usage for the values.

For many elements participants are given the option to avoid providing an answer to the element. A participant may choose not to answer the question due to a preference or if they find the question to not be relevant to them. In order to support such an optional data collection mode, many elements in AoU include a permissible value of 'prefer not to answer' or 'skip'. We quantified how often the ability to avoid answering is available and how commonly it is used.

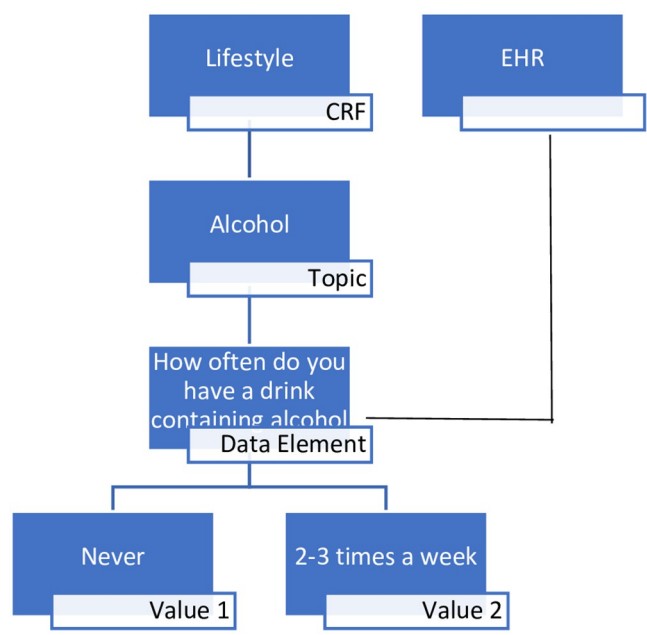

**Fig 1. Example data element hierarchy.**

Similar to the element level, we identified whether values belong to a standard terminology (e.g., LOINC or SNOMED CT) or were created custom for AoU and belong to the AoU created PPI terminology. We compared the element terminology with the terminologies used for its values to identify differences in terminology usage between the two levels (element and value). We identified when only one value terminology was used or if multiple value terminologies were needed. This included identifying UDEs that use only custom values, partial custom values, or fully standardized values. We also looked at CDEs and whether a custom value(s) was needed (and created) to fill any gaps that only using standard values would have created. An example of such an instance is the data element 'Total combined household income range in last year' (LOINC code:77244–2; OMOP concept_id: 46235933) which has 10 custom values from the PPI terminology including 'Annual Income: 50k-75k' (OMOP concept_id: 1585380).

### 3.3 Analysis of CRF and EHR data overlap

Along with analyzing the overall elements from CRF data, we further analyzed which elements crossover with the imported EHR data and are captured by both data sources. We identified these crossover elements (included in both CRF and EHR) by using the src_id. An element was considered from both sources if it had data rows from both the CRF source (PM/PPI) and one of the EHR sites. For the crossover elements we calculated what percentage of the data for each element originates from each data source. This crossover commonly shows elements that are originally answered via CRF and updated via EHR importing.

## 4 Results

Complete results, the code used for the analysis and additional analyses for the study are available on the study GitHub repository at https://github.com/lhncbc/CRI/tree/master/AoU/CDE.

### 4.1 Research data elements from Case Report Forms

For a population of 329 070 participants, we found 1 033 elements were used for collecting data through CRFs in the AoU workbench. From these 1 033 elements, 15.9% (164 elements) are CDEs as they originate from an existing terminology. In terms of terminology and data element standardization, 103 elements are LOINC concepts (10.0% of the total elements), 60 are SNOMED CT concepts (5.8%%) and one is a UCUM concept (0.1%). The remaining 84.1% (869 elements) are unique to AoU, from the PPI terminology, and we consider these UDEs. Table 2 shows examples of CRF elements and the event counts for each element. The complete list of CRF elements can be found in supplemental file S1-AOU_CRF_elements at the project repository (https://github.com/lhncbc/CRI/tree/master/AoU/CDE).

All 60 of the SNOMED CT elements are related to family or personal medical history, such as 'Family history of sickle cell anemia' (OMOP concept_id: 4050803; SNOMED CT concept: 160320002). The only UCUM concept is 'Days per week?' (OMOP concept_id: 8621) which is a dependent element branched off other elements on a CRF based on the answer of the other element. For example, if a participant answers 'Yes' to the question 'During the last 7 days, did you do vigorous physical activities like heavy lifting, digging, aerobics, or fast bicycling?' (OMOP concept_id: 1333286), the question 'Days per week?' will then be answered.

In terms of data type, the vast majority (96.0%, 992 out of all 1 033 CRF elements) are categorical, while only 41 (4.0%) are numeric. The categorical elements allow only a finite set of permissible values as answers. Permissible values primarily use OMOP concepts listed as value_concept_ids to express responses, while the numeric elements have a numeric value listed as value_as_number (possibly along with a value_as_concept_id).

**Table 2. Subset of CRF elements (ordered by descending event count).**

| Concept ID | Concept Name | Vocabulary ID | Concept Code | Event Count | CRF* |
|---|---|---|---|---|---|
| 3027018 | Heart rate | LOINC | 8867–4 | 13 609 305 | |
| 3025315 | Body weight | LOINC | 29463–7 | 3 215 985 | |
| 40766240 | Are you covered by health insurance or some other kind of health care plan [PhenX] | LOINC | 63513–6 | 931 119 | The Basics |
| 1333104 | In the past month, to cope with social distancing and isolation, are you doing any of the following? Select all that apply. | PPI | | 918 790 | COPE |
| 4214956 | History of clinical finding in subject | SNOMED | 417662000 | 597 652 | |
| 1585636 | Recreational Drug Use: Which Drugs Used | PPI | | 541 557 | Lifestyle |
| 40771103 | How often do you have a drink containing alcohol | LOINC | 68518–0 | 402 030 | Lifestyle |
| 40771090 | Current occupational status [SAMHSA] | LOINC | 68505–7 | 344 088 | The Basics |
| 1585389 | Health Insurance: Health Insurance Type | PPI | | 339977 | |
| 1586140 | Race: What Race Ethnicity | PPI | | 336 278 | The Basics |
| 1585370 | Home Own: Current Home Own | PPI | | 329 038 | The Basics |
| 1585766 | Overall Health: Medical Form Confidence | PPI | | 329 038 | Overall Health |

*A blank cell in the CRF column indicates data elements originating from a research visit instead of a specific CRF

**4.1.1 Analysis of initiative CDEs.** Of the 164 CDEs, 87 (53.1%) are initiative CDEs originating from 15 different data collection initiatives, with multiple elements from nine different initiatives. This includes 17 elements (19.5%) from PhenX which had the most initiative CDEs and 15 elements (17.2%) from PROMIS which had the second most initiative CDEs. Table 3 shows the counts of initiative CDEs and which initiative they originate from.

Some of the most commonly answered initiative CDEs were the PhenX elements 'Are you covered by health insurance or some other kind of health care plan' (OMOP concept_id: 40766240; LOINC code: 63513–6) and 'Current State' (OMOP concept_id: 40766229; LOINC code: 63501–1), which refers to the geographic state the participant is located. Along with the two mentioned PhenX CDEs, the PROMIS element 'In general, how would you rate your physical health' (OMOP concept_id: 40764340; LOINC code: 61579–9) was also one of the most commonly answered initiative CDEs. 15 initiative CDEs, including the three listed above, were answered by more than 99.9% of participants. Table 4 lists these commonly answered initiative CDEs.

**4.1.2 Elements by CRF.** The three core CRFs (The Basics, Overall Health, and Lifestyle) contain 44 (4.3%) elements available at the registered tier, while the additional surveys

**Table 3. Count of CDEs from each initiative.**

| Initiative | Element Count | Percent of Initiative CDEs |
|---|---|---|
| PhenX | 17 | 19.54% |
| PROMIS | 15 | 17.24% |
| Reported.PHQ | 14 | 16.09% |
| Perceived Stress Scale-10 | 10 | 11.49% |
| MOS Social Support Survey | 9 | 10.34% |
| HHS.ACA Section 4302 | 6 | 6.90% |
| SAMHSA | 5 | 5.75% |
| IPAQ | 3 | 3.45% |
| UCLA Loneliness Scale v3 | 2 | 2.30% |

**Table 4. Initiative CDEs answered by more than 99.9% of participants.**

| Concept ID | Concept Name | LOINC Code | Event Count | Initiative |
|---|---|---|---|---|
| 40766240 | Are you covered by health insurance or some other kind of health care plan [PhenX] | 63513–6 | 931 119 | PhenX |
| 40766229 | Current state [PhenX] | 63501–1 | 329 039 | PhenX |
| 40764338 | In general, would you say your health is [PROMIS] | 61577–3 | 329 038 | PROMIS |
| 40764339 | In general, would you say your quality of life is [PROMIS] | 61578–1 | 329 038 | PROMIS |
| 40764340 | In general, how would you rate your physical health [PROMIS] | 61579–9 | 329 038 | PROMIS |
| 40764341 | In general, how would you rate your mental health, including your mood and your ability to think [PROMIS] | 61580–7 | 329 038 | PROMIS |
| 40764342 | In general, how would you rate your satisfaction with you social activities and relationships [PROMIS] | 61581–5 | 329 038 | PROMIS |
| 40764343 | To what extent are you able to carry out your everyday physical activities such as walking, climbing stairs, carrying groceries, or moving a chair [PROMIS] | 61582–3 | 329 038 | PROMIS |
| 40764344 | How would you rate your pain on average in past 7 days [PROMIS] | 61583–1 | 329 038 | PROMIS |
| 40764345 | How would you rate your fatigue on average in past 7 days [PROMIS] | 61584–9 | 329 038 | PROMIS |
| 40766357 | In your entire life, have you had at least 1 drink of any kind of alcohol, not counting small tastes or sips [AUDADIS-IV] | 63633–2 | 329 038 | AUDADIS-IV |
| 40771090 | Current occupational status [SAMHSA] | 68505–7 | 344 088 | SAMHSA |
| 40771091 | What is the highest grade or level of schooling you completed [SAMHSA] | 68506–5 | 329 038 | SAMHSA |

(Personal Medical History, Healthcare Access & Utilization, COVID-19 Participant Experience and Family History) account for most of the elements with 642 (62.1%). GROR and the three consent forms (Primary Consent Update, Consent PII, and EHR Consent PII) are separate from the core and additional CRFs as they are associated strictly with consent and genetic information rather than personal or health information. GROR refers to a questionnaire regarding consent and participant protections around DNA testing. These four forms account for 41 (4.0%) elements. A total of 306 (29.6%) elements are not linked to any specific CRF and are likely generated during the research visit. Examples of such elements are 'Heart Rate' or 'Current State'. An additional 693 elements from CRFs were either redacted from the workbench in the registered tier of access or did not have enough data available to report on in accordance with AoU's data privacy policies (less than 20 instances). These elements were not present in the analyzed data and are not reported on. Table 5 shows the number of elements, CDEs, and UDEs for each CRF.

**Table 5. Count of elements, CDEs and UDEs by CRF.**

| Case Report Form (CRF) | Elements | Percent of Total Elements | UDEs | CDEs |
|---|---|---|---|---|
| Personal Medical History | 461 | 44.63% | 461 | 0 |
| No CRF Declared | 306 | 29.62% | 0 | 306 |
| COVID-19 Participant Experience (COPE) | 118 | 11.42% | 118 | 0 |
| Healthcare Access & Utilization | 57 | 5.52% | 57 | 0 |
| GROR | 26 | 2.52% | 26 | 0 |
| **The Basics** | **21** | **2.03%** | **9** | **12** |
| **Lifestyle** | **14** | **1.36%** | **4** | **10** |
| Primary Consent Update | 10 | 0.97% | 10 | 0 |
| **Overall Health** | **9** | **0.87%** | **8** | **1** |
| Family History | 6 | 0.58% | 6 | 0 |
| Consent PII | 3 | 0.29% | 3 | 0 |
| EHRConsent PII | 2 | 0.19% | 2 | 0 |

Only the three core CRFs (The Basics, Overall Health and Lifestyle [bolded in Table 5]) included CDEs. Of the three, The Basics and Lifestyle had more than half of the elements being CDEs, while Overall Health only included one CDE (of eight total elements). Of the core and additional CRFs, the CRF with the most elements is Personal Medical History with 461 elements. Family History had the fewest with six elements.

## 4.2 Permissible values

**4.2.1 Value overview.** On a permissible value level, we found 932 distinct values for CRF elements as a whole and 4 592 element-value combinations. Most values are only used for a single element, 672 values (72.1%), compared to 260 values (27.9%) which are used for multiple elements. The most used values (aside from 'Skip') were 'No' (used for 349 elements, with 9 214 901 instances) and 'Yes' (241 elements with 3 949 815 instances). Both values originate from the LOINC terminology. Table 6 shows examples of element-value combinations including the number of total responses, the number of responses by value chosen and the percentage of responses that were a specific value. The complete results available at the project repository in file S2_AOU_CRF_values.xlsx (https://github.com/lhncbc/CRI/tree/master/AoU/CDE).

**Table 6. Examples of CRF element-value combinations.**

| Element | Element Vocabulary ID | Element Count | Value | Value Vocabulary ID | Value Count | Percent of Element Values |
|---|---|---|---|---|---|---|
| Are you covered by health insurance or some other kind of health care plan [PhenX] | LOINC | 931119 | Yes | LOINC | 554459 | 59.5% |
| | | | Insurance through a current or former employer or | PPI | 136606 | 14.7% |
| | | | Medicare, for people 65 and older or people with certain disabilities | PPI | 110455 | 11.9% |
| | | | Insurance purchased directly from an insurance company | PPI | 42737 | 4.6% |
| | | | No | SNOMED | 27820 | 3.0% |
| | | | Medicaid, Medical Assistance, or any kind of government-assistance plan for those with low incomes or disability | PPI | 21589 | 2.3% |
| | | | Any other type of health insurance or health coverage plan | PPI | 12501 | 1.3% |
| | | | Veterans Affairs (VA) (including those who have ever used or enrolled for VA health care) | PPI | 11955 | 1.3% |
| | | | PMI: Skip | PPI | 9429 | 1.0% |
| | | | Don't know | LOINC | 3270 | 0.4% |
| How often do you have a drink containing alcohol | LOINC | 402030 | Monthly or less | LOINC | 117229 | 29.2% |
| | | | Never | LOINC | 81020 | 20.2% |
| | | | 2–4 times a month | LOINC | 78459 | 19.5% |
| | | | 2–3 times a week | LOINC | 58297 | 14.5% |
| | | | 4 or more times a week | LOINC | 57085 | 14.2% |
| | | | I prefer not to answer | LOINC | 9940 | 2.4% |
| Home Own: Current Home Own | PPI | 329038 | Current Home Own: Own | PPI | 145365 | 44.2% |
| | | | Current Home Own: Rent | PPI | 133659 | 40.6% |
| | | | Current Home Own: Other Arrangement | PPI | 35481 | 10.8% |
| | | | I prefer not to answer | LOINC | 14419 | 4.4% |

Table 7. Terminology usage for the included values.

| Vocabulary ID | Distinct Values | Percentage of Values | Elements with value terminology | Percent of Elements with values terminology |
|---|---|---|---|---|
| LOINC | 194 | 20.82% | 748 | 72.41% |
| PPI | 344 | 36.91% | 768 | 74.35% |
| SNOMED CT | 357 | 38.30% | 338 | 32.72% |
| AoU_General | 13 | 1.39% | 12 | 1.16% |
| None | 1 | 0.11% | 68 | 6.58% |
| ICD10CM | 9 | 0.97% | 1 | 0.10% |
| ICD9CM | 11 | 1.18% | 1 | 0.10% |
| UCUM | 3 | 0.32% | 2 | 0.19% |

On average, each categorical element had 4.5 permissible values with a median of three values per element. The maximum number of values for an element was 59 (for element 'History of clinical finding'; OMOP concept_id: 4214956).

**4.2.2 Answer avoidance.** We also found that the majority of CRF elements included an option for answer avoidance through the value 'PMI: Skip' (OMOP concept_id: 903096). Of the 1 033 elements, 748 (72.4%) Included an option to skip the question. Overall, there were a total of 8 236 129 instances where the custom value of 'PMI: Skip' was used, which made it the second most used value concept overall. Additionally, the LOINC concept 'I prefer not to answer' (OMOP concept_id: 1177221, LOINC code: LA29631-1) was used 16 5456 times for 63 elements. Answer avoidance by using one of these two concepts was found as an option for 801 (77.1% of 1 033) elements.

**4.2.3 Value terminologies.** Most values from CRF elements in AoU are common standardized values previously developed in an existing terminology such as LOINC or SNOMED CT. Of the 932 distinct values, 575 (61.7%) were part of a previously existing terminology. Of the 575 standardized values, most are from SNOMED CT (357, 62.1%) or LOINC (194, 33.9%). 357 (38.3%) of the values are unique, developed by AoU and are part of the created PPI (or AoU General) terminology. Table 7 shows which terminologies are used to represent values, the total number of values that come from each terminology, the percentage of all values, and the number of distinct elements that have values from a given terminology.

On an element level, 63 (6.1%) had all values from the same terminology as the element, while 186 (18.0%) had values completely from a different terminology than that of the element. In addition, 784 (75.9%) elements had values from multiple terminologies.

We found that most of the custom element concepts (UDEs) at least partially used standardized concepts for values. 819 (94.2% of 869) UDEs included values from standardized terminologies, compared to 50 UDEs that only included custom values. Furthermore, 239 (27.5%) of the UDEs only use standardized values. For example, the custom concept 'Cancer: How Old Were You When You Were First Told You Had Esophageal Cancer' (OMOP concept_id: 43530449) had all three values from LOINC ('Adult', 'Mature Adult 65–74 years', and '75+ years'). Alternatively, many of the custom values are used with CDEs to fill any gaps only using existing values may leave. Analyzing element-value combinations, we found 145 (88.4%) CDEs included at least one custom value.

## 4.3 Data elements included in both CRFs and EHR

When reviewing the included real world EHR data in conjunction with the research CRF data we found 64 elements were included in both. Some of the elements, such as 'Body Weight' and

**Table 8. Elements included in both EHR and CRF data.**

| Element Concept ID | Element | Element Vocabulary ID | Element Concept Code | Event Count | Percent of Total Participants | Percent of Events from CRFs |
|---|---|---|---|---|---|---|
| 3027018 | Heart rate | LOINC | 8867–4 | 13 609 305 | 82.09% | 5.90% |
| 3012888 | Diastolic blood pressure | LOINC | 8462–4 | 9 758 808 | 81.85% | 8.15% |
| 3038553 | Body mass index (BMI) [Ratio] | LOINC | 39156–5 | 2 421 692 | 80.52% | 10.89% |
| 4214956 | History of clinical finding in subject | SNOMED | 417662000 | 597 652 | 45.63% | 29.81% |
| 40771103 | How often do you have a drink containing alcohol | LOINC | 68518–0 | 402 030 | 86.21% | 99.06% |
| 40771104 | How many standard drinks containing alcohol do you have on a typical day [SAMHSA] | LOINC | 68519–8 | 317 050 | 69.87% | 98.83% |
| 40766929 | How many cigarettes do you smoke per day now [PhenX] | LOINC | 64218–1 | 127 781 | 37.76% | 97.18% |
| 4179963 | Family history of breast cancer | SNOMED | 429740004 | 51 401 | 3.91% | 5.32% |

'Diastolic Blood Pressure', were generated from the research visit rather than an actual survey CRF.

The amount of data from participant completed CRFs compared to encounters recorded in the EHR varies greatly depending on the element. For example, for the question, 'How often do you have a drink containing alcohol' (LOINC code: 68518–0; OMOP concept_id: 40771103), 99.1% of the data recorded comes from events recorded in CRFs. In contrast, the element 'Heart Rate' (LOINC code: 8867–4; OMOP concept_id: 3027018) has 94.1% of the data reported through events from EHR import. Table 8 shows a subset of EHR-CRF crossover elements. For the complete list of these such elements see our project repository (S3_CRF-EHR-crossover.xlsx, https://github.com/lhncbc/CRI/tree/master/AoU/CDE).

## 5 Discussion

The AoU program adopted a very modern approach to research data representation with their decision to adopt a common data model. The use of a CDM (OMOP CDM) allows researchers already familiar with this model to start utilizing the data in a shorter time. In contrast, a similar research program in the UK called UK Biobank uses a non-standard, study-specific format. Researchers must first spend time learning this custom format. The same is true for working with data from Framingham study data (available via dbGaP) [11].

Specifically, by using the OMOP CDM, AoU allows for the easier use of previously created tools developed by the OHDSI community. Such tools include the Atlas cohort tool (to which we developed a function in our r4aou package that uses Atlas cohorts [called aou_execute_cohort]) [9, 12] and Achilles for dataset characterization [13]. These tools can be used in conjunction with AoU due to the use of the OMOP CDM as opposed to the need for researchers to redevelop such tools for each unique data platform.

### 5.1 EHR and research data integration

The decision to adopt a CDM may have been triggered by the need to accommodate the import of EHR data and the seamless integration of this data with research-specific data collection. AoU uses EHR imported data to complement the data shown about an individual from research CRFs. While the research CRFs included only 1 033 elements, when combined with imported EHR data the number of total elements rises to 10 792. The inclusion of EHR linked data allows for more information about a participant than just what is collected through the typical research process, while also allowing for the potential monitoring of changes in a

participant's lifestyle or clinical background that are found through imported EHR data, rather than through having to repeat the research process. An example of this can be seen with cigarette smoking quantification (OMOP concept_id: 40766929) which may be established by the research questionnaire (in 97.2% of instances) but later updated via EHR import (in 2.8% of instances). Since this element is collected by both research and EHR questionnaires, changes in lifestyle can be detected. However, it is necessary to consider the potential of bias in the monitoring of such changes. The lack of direct follow-up through survey repetition and instead the reliance on EHR data for the monitoring of lifestyle changes may bias the data towards participants who more regularly use the healthcare system either due to needs based clinical backgrounds, healthcare access or health consciousness. The presence of a more limited participant population in EHR records leaves the possibility that lifestyle changes for the EHR population may not be representative of the population as a whole, which would otherwise be discoverable through survey repetition. To this point, only 59.6% of participants have available EHR data to complement their research data and in total 64 data elements are collected via both research CRFs and EHR import.

By the volume of data and the volume of distinct data elements, the routine healthcare context (EHR data) vastly outnumbers the research context (9.5 times as many elements in EHR than CRF, and five times as many instances). While some may regard the OMOP model to be a predominantly EHR data format, AoU shared data proves that OMOP is sufficiently flexible to accommodate study specific, research UDEs. Moreover, the terminology layer of the OMOP concept can equally well accommodate concept relationships within the research-specific PPI terminology.

## 5.2 CDE usage

AoU data demonstrates that 164 CRF data elements could be represented using routine healthcare terminologies (LOINC, SNOMED CT, etc.). While AoU was able to include a set of CDEs (164 elements), the fact that 15.9% of the elements are CDEs and that all CDEs originated from just three different CRFs leaves some room for potential increased integration of established CDEs. However, this percentage of CDEs used by AoU can be further augmented when considering the fact that 61.7% of values come from an established terminology. This use of common values was highly visible in some CRFs such as Personal Medical History where SNOMED CT was mostly used to express the conditions a participant was diagnosed with. For example, a UDE 'Digestive: Ulcerative Colitis Currently' (OMOP concept_id: 43530328) was used but has a value of Ulcerative Colitis (OMOP concept_id: 81893; SNOMED CT: 64766004) which is a common value to express the diagnosis of a condition and can be used much in the same way a CDE would be used. Including such CDEs and common values makes it easier to use existing tools and compare analyses of equivalent elements.

## 5.3 Types of CDEs

A closer look at LOINC encoded CDEs shows that LOINC CDEs originate from the inclusion into LOINC of data elements from established data collection instruments included in various initiatives such as PhenX, Patient Health Questionnaire (PHQ), or PROMIS (initiative CDEs).

The decision to adopt the OMOP model has significantly shaped how data elements are standardized. However, how to refer to CDEs in various initiatives that define them is an ongoing clinical research informatics challenge. A modern view of this challenge can classify data elements along two axes (by origin and by presence in routine healthcare terminology). These two axes overlap and produce 4 distinct classes. The *origin* axis splits CDEs into O1: *routine healthcare* elements that are collected in EHRs during regular clinical encounters and O2:

Table 9. Data element classification and element count by origin and terminology axes.

|  | T1: *ingested*: 164 | T2: *non-ingested*: 869 |
|---|---|---|
| **O1 Healthcare: 9759** | O1-healthcare-T1-ingested:64 | N/A |
| **O2-research: 1033** | O2-research-T1-ingested:100 | O2-resarch-T2-non-ingested: 0 |

*research initiative* elements defined by research standards initiatives, such as PROMIS or PhenX.

Further division occurs, with the presence-in-routine-healthcare-terminology axis splitting CDEs into T1: *ingested* elements that are included in a routine healthcare terminology (given some dataset production date) and T2: *non-ingested* elements that in judgement of Standard Development Organizations (SDO) overseeing routine healthcare terminologies did not reach criteria for ingestion. To further simplify the division, we use short terms healthcare and research for origin axis and short terms ingested and non-ingested for terminology axis. Note that UDEs (elements unique to a single study) still reside outside of this classification. The classification is only meant for standardized and common elements.

Using this classification, we can characterize AoU data elements as relying on 164 ingested CDEs and 869 study-specific UDEs. The research and healthcare overlap data shows that 64 of those 164 ingested CDEs are routine (type: O1-healthcare-T1-ingested) and the remaining 100 are research (type O2-research-T1-ingested). Table 9 shows these axes and the counts of elements by classification.

As Table 9 shows not all elements are present on these axes as only CDEs included in the study data are present in this classification, excluding study specific data elements and routine healthcare data elements not included in the study. There are no CDEs in class of O2-resarch-T2-non-ingested. That is because the OMOP model terminology layer as of 2021 mostly includes routine healthcare terminologies and not very many research harmonization initiatives. A potential reason for this absence may lie in possibly poor distribution formats of CDEs created by research harmonization initiatives and a lack of pressure to enforce use of CDE identifiers by harmonization initiatives or other research stakeholders.

The 64 overlapping CDEs demonstrate penetration of initially research-only elements into routine healthcare, e.g., a mental health screening question (OMOP concept_id: 3042924; LOINC code: 44250–9; 'Little interest or pleasure in doing things in last 2 weeks') originating from Patient Health Questionnaire (PHQ). Ingestion into routine healthcare terminologies may sometimes not be desired by CDE initiatives. Some may argue that this ingestion of research data elements renders the initiative redundant, as users now have an option to obtain these data elements from outside of the original source. They may also point to incomplete ingestion. The terminology model of LOINC or SNOMED CT may not align with the information model used by the CDE initiative (e.g., PhenX elements obtained directly from the PhenX website with all metadata may not align with LOINC terminology in a shortened format). Others may view ingestion as a gateway to a possible wider adoption of the data element (if ingestion is done properly). Copyright issues may also exist and are typically retained by the initiative while the terminology SDO is doing the ingestion with formally granted permission. In the case of PROMIS, only a limited portion of research CDEs are ingested, and the full set requires a payment and research or a commercial license.

### 5.4 Answer avoidance

The analysis of skipped answers shows that a patient facing research questionnaire may have a valid answer of 'prefer not to answer'. A total of 77.1% of CRF elements allowed for avoiding

the question through the choice of 'skip' or 'prefer not to answer'. Some terminologies such as SNOMED CT do not provide a way to represent this answer. In contrast, the FHIR standard provides 'asked-declined' value in data-absent-reason value sets [14].

## 5.5 AoU development and adoption of OMOP

We are the first to analyze the full AoU dataset and its usage of CDEs, as it is only four years old and is still growing and being added to in terms of both participants and data elements. While the goal of the program is to enroll one million participants, participant level data has been made available to researchers very early in the program, allowing for efficient research practices. The ability to access participant level data through the researcher workbench allows for valuable insights into the data elements being collected.

AoU chose not to utilize the OMOP SURVEY_CONDUCT table that was designed to store an instance of a completed survey or questionnaire (see https://ohdsi.github.io/CommonDataModel/cdm60.html#survey_conduct). If the table was used, it would allow for the accounting of which CRF was used (survey_concept_id), whether the survey was completed with assistance or independently (assisted_concept_id) and how long it took to complete the CRF.

## 5.6 Limitations

Our research was limited as it only includes concepts available in the registered tier (as the controlled tier was unavailable at the beginning of our analysis) of access and with at least 20 participant responses and excludes certain CRF elements that may be redacted from this tier of access. In addition, AoU adopted some elements from other data collection initiatives as seen at the bottom of the survey document under source [15], however we consider these UDEs as the elements are formalized in the custom PPI terminology and not originating from one of the established terminologies in Athena. Another limitation is when assessing the crossover of CRF and EHR elements we did not do any semantic mapping and instead relied on the EHR sites using the same OMOP concepts used for the CRF elements. This would also be limited by the heterogeneity of each site, as each site may use different concepts or provide limited amounts of measurement or observation data. We also did not confirm the accuracy of the mapping of elements in the surveys to OMOP concept_ids, however, because the OMOP model preserves the original source values, it is possible to re-inspect and re-evaluate the mapping at the time of analysis. For the assigning of elements to CRFs our work is limited as we had to develop our own association dictionary from the available Athena CONCEPT_RELATIONSHIP table rather than a provided association dictionary from AoU.

## 6 Conclusion

CDEs represent an effort to standardize data collection across human clinical studies that allows for easier analysis and understanding of collected data. The AoU program adopted a modern approach to research data representation by adopting the OMOP CDM and complemented its usage by including a set of CDEs and established values. AoU included 1 033 distinct elements, 932 distinct values and 4 592 element-value combinations for the set of CRFs in the program. 15.9% (164 elements) of the elements were CDEs, with many of the CDEs (87 CDEs; 53.1% of all CDEs) coming from a previously used CDE imitative. 61.7% (576 distinct values) of distinct values are common as well and come from an established terminology. The use of such CDEs and established values allows for a more efficient analysis process of the collected data and the improved usage of previously established analytic tools. Increased CDE usage in such large studies (like AoU) would better facilitate the collection and useability of the

data generated by the study and should be considered by other studies during study development and data collection.

## Acknowledgments

We would like to thank Nick Williams, James Mork and Kerry Goetz for providing comments on drafts of this manuscript. We would also like to thank the All of Us program. The *All of Us* Research Program is supported by the National Institutes of Health, Office of the Director: Regional Medical Centers: 1 OT2 OD026549; 1 OT2 OD026554; 1 OT2 OD026557; 1 OT2 OD026556; 1 OT2 OD026550; 1 OT2 OD 026552; 1 OT2 OD026553; 1 OT2 OD026548; 1 OT2 OD026551; 1 OT2 OD026555; IAA #: AOD 16037; Federally Qualified Health Centers: HHSN 263201600085U; Data and Research Center: 5 U2C OD023196; Biobank: 1 U24 OD023121; The Participant Center: U24 OD023176; Participant Technology Systems Center: 1 U24 OD023163; Communications and Engagement: 3 OT2 OD023205; 3 OT2 OD023206; and Community Partners: 1 OT2 OD025277; 3 OT2 OD025315; 1 OT2 OD025337; 1 OT2 OD025276. In addition, the All of Us Research Program would not be possible without the partnership of its participants.

## Author Contributions

**Conceptualization:** Craig S. Mayer, Vojtech Huser.

**Data curation:** Craig S. Mayer.

**Formal analysis:** Craig S. Mayer.

**Funding acquisition:** Vojtech Huser.

**Investigation:** Craig S. Mayer, Vojtech Huser.

**Methodology:** Craig S. Mayer, Vojtech Huser.

**Project administration:** Craig S. Mayer, Vojtech Huser.

**Resources:** Craig S. Mayer, Vojtech Huser.

**Supervision:** Vojtech Huser.

**Validation:** Craig S. Mayer.

**Writing – original draft:** Craig S. Mayer, Vojtech Huser.

**Writing – review & editing:** Craig S. Mayer, Vojtech Huser.

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
