## [Decision Letter · Decision Letter 0]

9 Feb 2023

PONE-D-21-40725Learning important common data elements from shared study data: the All of Us program analysisPLOS ONE

Dear Dr. Mayer,

Thank you for submitting your manuscript to PLOS ONE. After careful consideration, we feel that it has merit but does not fully meet PLOS ONE’s publication criteria as it currently stands. Therefore, we invite you to submit a revised version of the manuscript that addresses the points raised during the review process.

Please see attached comments from two reviewers attached. While reviewer one confirms that the work is a valuable addition to the literature I would suggest focusing on addressing the feedback from reviewer two. We thank you for your time and hope that you choose to submit a revision given the reviews provided.

We look forward to receiving your revised manuscript.

Kind regards,

Jake Michael Pry, PhD, MPH

Academic Editor

PLOS ONE

Journal Requirements:

Reviewers' comments:

Reviewer's Responses to Questions

**Comments to the Author**

1. Is the manuscript technically sound, and do the data support the conclusions?

Reviewer #1: Yes

Reviewer #2: Yes

2. Has the statistical analysis been performed appropriately and rigorously? 

Reviewer #1: N/A

Reviewer #2: N/A

3. Have the authors made all data underlying the findings in their manuscript fully available?

Reviewer #1: Yes

Reviewer #2: Yes

4. Is the manuscript presented in an intelligible fashion and written in standard English?

Reviewer #1: Yes

Reviewer #2: Yes

5. Review Comments to the Author

Reviewer #1: The authors have examined the common data elements from the All of Us program data. The data is available at our project repository

https://github.com/lhncbc/CRI/tree/master/AoU/CDE

This publication is in a combination of present and past tense. Even though the All of US program maybe continuing to enroll participants, the data for this study was obtained at a specific timeline (late 2021), and the analyses performed retrospectively, and hence could be described in past tense.

Overall the paper is well written. Increased CDE usage in large studies like the All of Us program would better facilitate the usability of the data.

Reviewer #2: Summary

The authors analyze the All of US (AoU) program with respect to common data elements (CDEs) which are standardized across multiple clinical studies. AoU uses the OMOP Common Data Model for standardization. The authors investigate the frequency of CDEs as compared to unique data elements (UDEs), defined specifically for AoU. Among other details, the origin of the CDEs is described.

General

The manuscript deals with an important topic for the medical research infrastructure. I have a few comments but overall the manuscript merits publication.

Comments

1) Several typos and grammar errors. Please correct.

2) Abstract:

Abbreviation PPI not defined.

“originated from a previous data collection initiative, such as PhenX (17 CDEs) and PROMIS (15 CDEs)” plr: data collection initiatives

3) 2.1 All of Us program

Missing is the aim of the initiative and the planed follow-up. Is biomaterial collected by AoU? Is genetic data available? Please add.

4) The structures of the data elements and values are rather complex and not easy to grasp. Add figures which present the data structures and hierarchies and also add examples in the text.

5) Table 2: Why is the column “CRF” blank for some elements? Explain in legend to table 2.

6) Element ‘Current State’: Please explain what that means.

7) l 233: GROR: It is not explained what that is.

8) l 245: Brackets not consistent.

9) Table 6: Add the event counts. Why do the “Percent of Element Values” not add up to 100%? Please explain what the percent refers to. Please additionally use such a reference population that the percentages add up to 100%.

10) 5.1 EHR and research data integration:

“An example of this can be seen with cigarette smoking quantification (OMOP concept_id: 40766929) which may be established by research questionnaire (in 97.2% of instances) but later updated via EHR import (in 2.8% of instances).”

Discuss the possibility of bias because of the low follow-up percentages which might result in selective, biased information.

11) l. 388f: grammar error

12) 5.3 Types of CDEs:

Please illustrate the two axes and the number of elements contained in the four sets by a table or Venn diagram.

6. PLOS authors have the option to publish the peer review history of their article (what does this mean?). If published, this will include your full peer review and any attached files.

Reviewer #1: No

Reviewer #2: No

---

## [Author Response · Author response to Decision Letter 0]

16 Feb 2023

Dear Dr. Jake Michael Pry,

We thank you and the reviewers for the input and constructive comments on our manuscript. We have thoroughly reviewed each comment and our manuscript and have edited our paper taking into account each comment. We are now submitting the edited and improved version of our manuscript. 

See below our response (in bold font) to each reviewer comment. Our response follows the reviewer’s comments (shown in regular un-bolded text). 

Thank you, 

Craig Mayer and Vojtech Huser

Reviewer #1

This publication is in a combination of present and past tense. Even though the All of US program maybe continuing to enroll participants, the data for this study was obtained at a specific timeline (late 2021), and the analyses performed retrospectively, and hence could be described in past tense.

Thank you for the comment. We have changed the portions where we discuss our analysis, results and conclusions, to conform to past tense. For the purposes of describing the All of Us program and their data collection practices we maintain the text in present tense since we are describing the ongoing methodology of the program itself. While our analysis and the results we found were done in the past and therefore in the past tense, the All of Us program is ongoing, and we feel should be described in present tense for that reason. 

Reviewer #2

1) Several typos and grammar errors. Please correct.

Thank you for your comments. We have corrected typos and grammar mistakes throughout the manuscript.

2) Abstract:

Abbreviation PPI not defined.

We have defined PPI in the Abstract where the acronym first appears.

“originated from a previous data collection initiative, such as PhenX (17 CDEs) and PROMIS (15 CDEs)” plr: data collection initiatives

We have made the term data collection initiative plural in the text of the Abstract to conform with the multiple initiatives discussed.

3) 2.1 All of Us program

Missing is the aim of the initiative and the planed follow-up. Is biomaterial collected by AoU? Is genetic data available? Please add.

We added the aims and a brief description of additional data types. However, it would be out of scope to further discuss these data types and process as the manuscript is focused on CRF and EHR data and the presence of common data elements. The addition of these data types beyond just mentioning them do not contribute to the conclusions as they are not part of the common data model structure. The further details regarding the All of Us program was added to this section (2.1).

4) The structures of the data elements and values are rather complex and not easy to grasp. Add figures which present the data structures and hierarchies and also add examples in the text.

Figure 1 along with an explanation was added to section 3.2 to clarify this process and structure.

5) Table 2: Why is the column “CRF” blank for some elements? Explain in legend to table 2.

The reason the CRF column is blank for some data elements is that these elements originate from a research visit instead of a CRF. This explanation was added as a footer to Table 2.

6) Element ‘Current State’: Please explain what that means.

To avoid initial confusion about the meaning of this data element, it was removed from Table 2. When it is mentioned later in the text in section 4.1.1, we explain that ‘Current State’ refers to the geographic location of the participant.

7) l 233: GROR: It is not explained what that is.

We added an explanation of what GROR refers to in section4.1.2. The exact meaning of the GROR acronym is not made apparent by the All of Us program or OMOP terminology. Instead, we explained the general purpose of the GROR CRF.

8) l 245: Brackets not consistent.

We fixed this with the addition of a close bracket for bracket agreement.

9) Table 6: Add the event counts. Why do the “Percent of Element Values” not add up to 100%? Please explain what the percent refers to. Please additionally use such a reference population that the percentages add up to 100%. 

Table 6 has been redone to include the count of total responses and responses by value chosen to describe the percentage. All values were included to see 100% of the population’s responses. A sentence was added before the table to explain the contents.

10) 5.1 EHR and research data integration:

“An example of this can be seen with cigarette smoking quantification (OMOP concept_id: 40766929) which may be established by research questionnaire (in 97.2% of instances) but later updated via EHR import (in 2.8% of instances).”

Discuss the possibility of bias because of the low follow-up percentages which might result in selective, biased information.

Sentences were added to section 5.1 to discuss this possibility. We have added sentences to this section describing the consideration of bias in the data due to the differences in population follow-up caused by the inclusion of EHR data instead of research process repetition.

11) l. 388f: grammar error

This sentence was edited to ensure proper grammar and clarify meaning.

12) 5.3 Types of CDEs:

Please illustrate the two axes and the number of elements contained in the four sets by a table or Venn diagram.

Table 9 was added to the section to better show this crossover and process. This should help clarify the premise described regarding the types of CDEs and the axes.

---

## [Decision Letter · Decision Letter 1]

13 Mar 2023

Learning important common data elements from shared study data: the All of Us program analysis

PONE-D-21-40725R1

Dear Dr. Mayer,

We’re pleased to inform you that your manuscript has been judged scientifically suitable for publication and will be formally accepted for publication once it meets all outstanding technical requirements.

Kind regards,

Jake Michael Pry, PhD, MPH

Academic Editor

PLOS ONE

Additional Editor Comments (optional):

Thank you for submitting to PLoS One, we hope that you will consider PLoS One for future work as well.

Reviewers' comments:

Reviewer's Responses to Questions

**Comments to the Author**

1. If the authors have adequately addressed your comments raised in a previous round of review and you feel that this manuscript is now acceptable for publication, you may indicate that here to bypass the “Comments to the Author” section, enter your conflict of interest statement in the “Confidential to Editor” section, and submit your "Accept" recommendation.

Reviewer #1: All comments have been addressed

Reviewer #2: All comments have been addressed

2. Is the manuscript technically sound, and do the data support the conclusions?

Reviewer #1: Yes

Reviewer #2: (No Response)

3. Has the statistical analysis been performed appropriately and rigorously? 

Reviewer #1: N/A

Reviewer #2: (No Response)

4. Have the authors made all data underlying the findings in their manuscript fully available?

Reviewer #1: Yes

Reviewer #2: (No Response)

5. Is the manuscript presented in an intelligible fashion and written in standard English?

Reviewer #1: Yes

Reviewer #2: (No Response)

6. Review Comments to the Author

Reviewer #1: All comments addressed. Overall the paper is well written. Increased CDE usage in large studies like the All of Us program would better facilitate the usability of the data.

Reviewer #2: (No Response)

7. PLOS authors have the option to publish the peer review history of their article (what does this mean?). If published, this will include your full peer review and any attached files.

Reviewer #1: No

Reviewer #2: No

---

## [Editor Report · Acceptance letter]

17 Mar 2023

PONE-D-21-40725R1 

Learning important common data elements from shared study data: the All of Us program analysis 

Dear Dr. Mayer:

I'm pleased to inform you that your manuscript has been deemed suitable for publication in PLOS ONE. Congratulations! Your manuscript is now with our production department. 

Kind regards, 

on behalf of

Dr. Jake Michael Pry 

Academic Editor

PLOS ONE